# Application of Laser Profilometry to Evaluation of the Surface of the Workpiece Machined by Abrasive Waterjet Technology

**Gerhard Mitaľ [1], Jozef Dobránsky [2],*[ID], Juraj Ružbarský [3] and Štefánia Olejárová [3]**

1   Department of Mechanical Engineering Technologies and Materials, Faculty of Mechanical Engineering, Technical University of Košice, Mäsiarska 74, 04001 Košice, Slovakia; gerhard.mital@tuke.sk
2   Department of Automotive and Manufacturing Technologies, Faculty of Manufacturing Technologies with a seat in Prešov, Technical University of Košice, Štúrova 31, 08001 Prešov, Slovakia
3   Department of Technical Systems Design and Monitoring, Faculty of Mechanical Engineering, Technical University of Košice, Štúrova 31, 08001 Prešov, Slovakia; juraj.ruzbarsky@tuke.sk (J.R.); stefania.olejarova@tuke.sk (Š.O.)
*   Correspondence: jozef.dobransky@tuke.sk; Tel.: +421-55-602-6350

**Abstract:** The paper is an evaluation of the surface roughness of various materials produced by water jet cutting (AWJ, abrasive water jet). A 3D laser profilometer developed at the Department of Design and Technical Systems Monitoring at our University was used in roughness measurement. To verify the values measured by the laser profilometer, another measurement was performed using a 2D contact roughness meter. The tests were done on aluminum and stainless-steel materials, respectively. Six samples were produced; three made of stainless steel and three made of aluminum. All samples were produced at a different feed rate of the cutting head. This was adapted to the different roughness required, per the manufacturer's material data sheets. Varying rates of separation translated into different qualities of the surfaces under evaluation. The evaluated roughness parameters were Ra and Rz. Dependencies were plotted in the chart based on the values measured, which were then compared and evaluated.

**Keywords:** roughness; profile; laser; non-contact; measurement; surface; parameters; technology

## 1. Introduction

Surface texture, which can be in many forms (i.e., tiny scratches, grooved structures produced by machining, etc.), is gradually gaining greater magnitude in the field of surface engineering [1].

Laser profilometry is one of the most recently developed approaches for surface roughness measurements, whilst pulsed thermography was used with the intention of associating the influence of surface roughness with thermal radiometry [2].

Interrupted laser microscopy and profilometry measurements of the friction surface show four evolution stages of the powder boundary layer development: localized formation, complete formation, local spalling, and large scuffing of the boundary layer [3].

The surface-emergent relief is not only the bearer of individual pieces of information, but also an image of the technology used to create it in the first place [4,5]. By evaluating the roughness parameters, it is possible to predict the sequence of operations required to achieve the desired surface quality and also to optimize the process of its creation [6].

Surface roughness is a parameter that often crops up in metal working processes. It is one of many factors that represent the quality of machining, which impacts productivity gains. From a mechanical oscillation point of view, various degrees of roughness are undesirable because they cause noise and

dynamic strain [7,8]. These result in fatigue and failure of the machine structure or its functional parts and a loss of energy or reduced performance. A chronological overview of optical profilometric systems, which are described as carrying out real time acquisition, digital signal processing and full-color 3D profile displays, have been introduced. The relevant operating principles, strengths and weaknesses of these devices have been discussed and discussions on major limitations and future challenges in high-speed optical profilometry have been accounted for [9,10].

Models describing the relationship between the machine and the workpiece and the surface quality of the latter began to appear in published papers several years ago. Z. Cojbasic, Petkovic, Shamshirband, Tong, Ch, Jankovic, Ducic, Baralic [11] defined several aspects of importance of the experimental analysis of surface roughness in the machining process. Unwanted surface roughness may indicate damage to functional parts of the machine, wear of tools, of the workpiece, the cutting head or other parts of the machine. These findings herald damage to, or lack of rigidity of the machine's structure, bearing components or the machine's other machining and separating parts [11].

Jeyapoovan and Murugan [12] have stated that the development of science and technology, and the application of the results of such developments, increase the importance of surface quality of parts. This largely affects their service life and reliability, and the operating accuracy, noise, corrosion and wear resistance, fatigue loss, or fatigue strength of components are largely quality-dependent [13,14]. The roughness value of surfaces that come into contact with each other is often a decisive factor [15]. It also significantly impacts the life, reliability and the operation of the technical equipment [16,17]. Liu, Lu, Yi, Wang and Ao [18] proposed a similar surface roughness measurement method to prove that the measurement process is largely influenced by the source of the laser light containing two colors, red and green, creating a relationship model between the overlapping index and roughness.

The work of C. Kang, H.X. Liu [19] and C. Lu [20] demonstrates that the surface profile and the workpiece surface roughness are the most important product quality characteristics, and, in most cases, they reflect important technical requirements for engineering components. Achieving the required surface quality is very important for the functional behavior of the part. In addition, the importance of the machined surface quality is constantly increasing, as are the demands placed on such surfaces [21,22]. Roughness represents the height of the unevenness as compared to the perfect and ideally smooth surface and emerges as a consequence of the work of a tool deployed [23] in machining and finishing [24]. The evaluated parameters were *Ra*, the mean arithmetic deviation of the examined profile, and *Rz*, the greatest height of the profile unevenness. The very first advantage of surface diagnostics by laser profilometry (LPM) (Figure 1) is that it also evaluates other surface characteristics such as *Rp*—the height of the largest profile projection, *Rq*—the mean quadratic deviation of the examined profile and *Rv*—the depth of the largest profile depression. These values are able to tell us more about the character of individual parts of the surface [25].

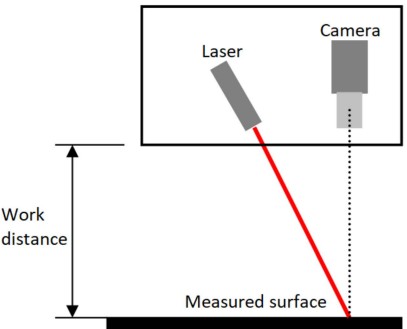

**Figure 1.** Measurement scheme using laser profilometer.

A fast surface profiling algorithm based on white light interferometry was designed using the sampling theory that was installed in the commercial system to reach the world's highest scanning

speed of 80 microns per second. The resolution of the system height is in the order of 10 nm for a measuring range greater than 100 microns [26,27].

The goals of the paper were postulated after careful review of the literature. The main objective of the experiment was to measure aluminum and stainless-steel samples using a laser profilometer and a contact roughness meter and to determine the effect the type of material had on the cutting speed. Data from the experimental measurements were processed into graphical dependencies and evaluated at the end of the paper. Another objective was to compare the measurement methods of the laser profilometer and the Mitutoyo SJ 400 roughness meter. The practical benefit of the experiment is, based on the analysis of the sample surfaces, the insight into whether the cutting machine is operating properly without any signs of impending failure. Comparing methods enabled us to determine their advantages and disadvantages, described at the end of the paper.

## 2. Methodology of the Research

The experiments carried out were aimed at monitoring differences in surface quality of different types of materials produced under the same material separation conditions, but with the cutting head speed adapted to the required roughness and material stiffness. Examined samples were made of aluminum and stainless steel and measured by contactless and contact methods. In the application of both methods, the sites of measurement were the same. The samples were measured and evaluated with the laser profilometer (contactless LPM, 2015, KVANT spol. s.r.o., Slovakia) and the Mitutoyo roughness tester (contact test, SJ400, Japan).

### 2.1. Preparation of Samples and Description of Tested Materials

Cutting device (AWJ, WJ4020 1Z-C0-PJ60 COBRA, PTV Prague was used, together with a PTV56-60 high pressure pump from FLOW SYSTEMS in Figure 2) was used for cutting samples at DRC, s.r.o. in Prešov.

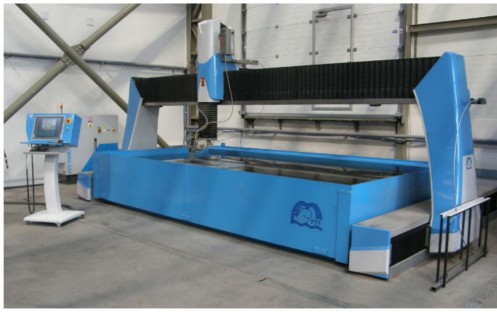

**Figure 2.** WJ4020 1Z-C0-PJ60 COBRA device.

In Table 1 are listed technical parameters of the equipment used for preparation of experimental samples.

**Table 1.** Technical parameters of the coordinate table WJ4020 1Z-C0-PJ60 COBRA.

| Parameter | Value |
| --- | --- |
| Maximum working speed | 20,000 mm·min$^{-1}$ |
| Positional accuracy | +/− 0.04 mm /300 mm |
| Repeatable accuracy | +/− 0.03 mm |
| Maximum speed | 30,000 mm·min$^{-1}$ |
| Z-axis travel | 500–700 mm |
| Number of Z-Supports | 1–2 |
| Number of cutting heads | 1–4 |
| Max. portal width | 4 m |
| Maximum length of linear guide | 30000 mm |

Analyzed materials were chosen on the base of the two basic requirements:

The usability and availability of materials in industry,
Different properties of analyzed material.

Sample (nomenclature) marking:

SS050—Stainless steel—cutting head feed rate (50 mm·min$^{-1}$)
SS120—Stainless steel—cutting head feed rate (120 mm·min$^{-1}$)
SS150—Stainless steel—cutting head feed rate (150 mm·min$^{-1}$)
AL120—Aluminum—cutting head feed rate (120 mm·min$^{-1}$)
AL220—Aluminum—cutting head feed rate (220 mm·min$^{-1}$)
AL370—Aluminum—cutting head feed rate (370 mm·min$^{-1}$)

Six samples were produced by the AWJ method (Figure 3). Three samples (AL120, AL220, AL370) were made of aluminum and three samples (SS050, SS150, SS150) were of stainless steel (SS—Stainless steel). For each sample, a different feed rate of the cutting head was used.

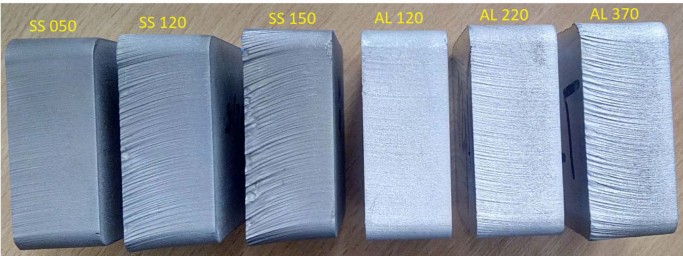

**Figure 3.** Experimental samples used.

For waterjet cutting, the company declares certain surface roughness to be achieved at different feed rates. The declared surface roughness values under different feed rates of the cutting head are shown in Table 2.

**Table 2.** Declared roughness of individual samples.

| Roughness | | |
|:---:|:---:|:---:|
| AL120 | AL220 | AL370 |
| 6.3 µm | 12.5 µm | 25 µm |
| SS050 | SS120 | SS150 |
| 25 µm | 12.5 µm | 6.3 µm |

Table 3 shows the parameters set for the AWJ cutting, constant for all samples in contrast to the varying speed rates of the cutting head.

**Table 3.** Cutting parameters setting.

| Parameter | Value |
|:---:|:---:|
| Pressure (MPa) | 379.21 |
| Type of abrasive | Australian grenade |
| Abrasive grain size | MESH 80 |
| Diameter of the water jet (mm) | 0.406 |
| Diameter of the guide tube (mm) | 0.889 |
| Amount of abrasive (g·min$^{-1}$) | 430 |

*2.2. Surface Measurement by Contactless Method*

The laser profilometer assembly (Figure 4) consists of basic and complementary parts. The basic parts include the support frame from components with vertical positioning of the measuring head and the programmable sample feed in the X and Y axes, the laser beam source, the lens and a CMOS (Complementary Metal Oxide Semiconductor) sensor camera. Additional parts include a computer with service and evaluation software and an image splitter.

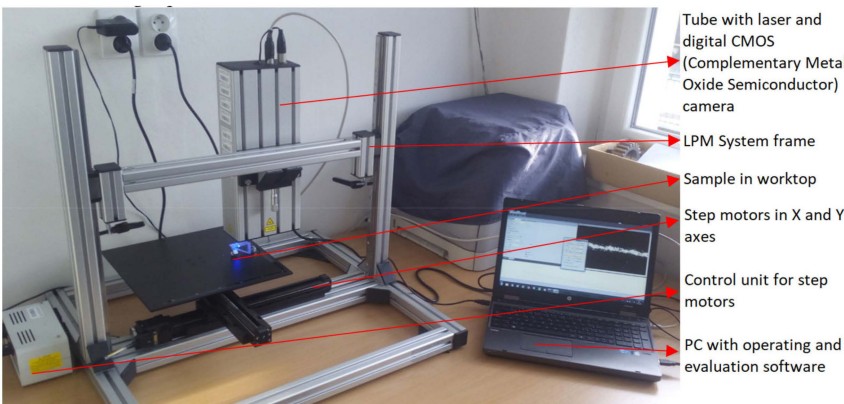

**Figure 4.** The laser profilometer assembly.

The optical part of the system consists of the AVT Marlin 131B camera and the Tamron 23FM50SP 50 mm lens with a visible area of 22 mm × 7 mm. The automated sample feed in the X and Y axes is achieved with the Standa 8MT160-300 stepper motors, up to a maximum of 300 mm in each axis. The system enables measurement of samples weighing no more than 8 kg with a positioning accuracy of 2.5 micrometers per step, each step consisting of 8 micrometers. The sensor resolution is 0.02 mm/pixel.

To eliminate the impact of vibrations on the measurement and on the quality of the measured data, the samples on the LPM worktop were placed in the plasticine.

Using the experimental system, it is possible to measure and evaluate the surface roughness parameters of samples according to ISO 11562 ($Rq$, $Rv$, $Rz$, $Ra$, $Rp$). The results of measured profile evaluation can be exported in csv format, suitable for further processing of experiments in the form of raw data.

The image splitter Matrox TripleHead2Go-Digital Edition delivers crisp processing and evaluation of the measured data images. This device serves the purpose of splitting one graphic output from the computer into three independent graphic outputs, and in combination with the use of another graphic output and the choice of an extended desktop, four independent images can be obtained by having a different part of the desktop on each monitor.

The Main Principle of the System

The system employs laser profilometry through a so-called triangulation principle (Figure 5), in which a laser line is projected onto the measured surface under a certain angle, which is subsequently scanned by a digital camera placed perpendicularly above the scanned surface.

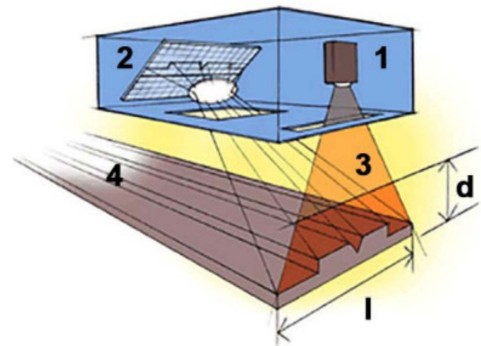

**Figure 5.** Principle of LPM measurement. (1) CMOS camera; (2) Laser light source; (3) Laser light on the surface; (4) Measured surface; (d) Working distance; (l) Measuring range.

## 2.3. Surface Measurement by Contact Method

As mentioned earlier in the introduction, for the verification and comparison of the measured values, the measurement of the samples was also carried out on the Mitutoyo SJ400 contact roughness tester (Figures 6 and 7). The technical parameters of the contact roughness tester are shown in the Table 4.

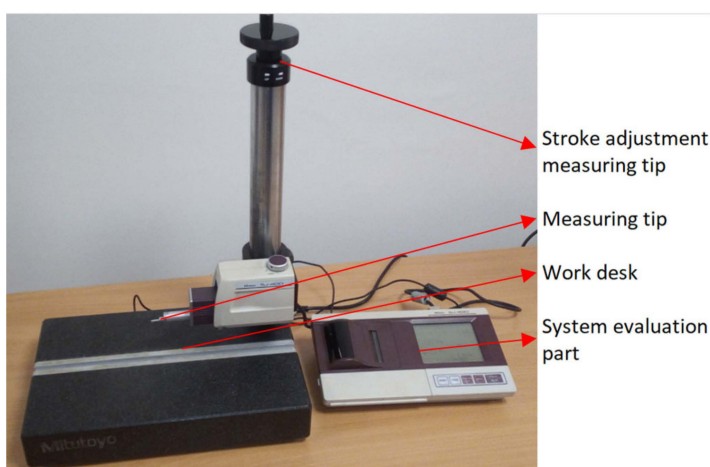

**Figure 6.** Contact method Mitutoyo SJ 400.

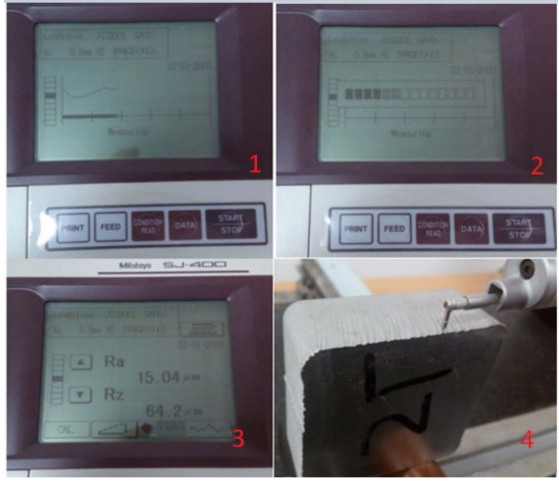

**Figure 7.** Mitutoyo SJ 400 software and hardware (1. Measuring phase, 2. Evaluation phase, 3. Measurement values, 4. Movement of the tip over the surface of the component).

**Table 4.** Technical parameters of the Mitutoyo SJ400 roughness tester.

| Parameter | Value |
|---|---|
| Measuring speed (mm·s$^{-1}$) | 0.05; 0.1; 0.5; 1.0 |
| Speed of return (mm·s$^{-1}$) | 0.5; 1.0; 2.0 |
| Positioning | ± 1.5° (inclination); 10 mm (up/down) |
| Measurement range/resolution (μm) | P (primary); R (roughness); W (filtered waviness) |
| Evaluated parameters | 0.889 |
| Digital filter | 2CR; PC75; Gauss |
| Cutoff length (mm) | 0.08; 0.25; 1.8; 2.5; 8 |

The same part of the measured surface was selected for each sample in both methods of measurement. The sample materials were defined as EN5083 (aluminum) and X5CrNi18-10 (A304 austenitic stainless steel) by the respective standard.

### 2.4. Experimental Sample Measurement

Quality measurement in terms of roughness was performed on two types of materials. The measured samples were of the same thickness and were made at different feed rates. Three samples were made from each material.

The Measurement Site of the Ra and Rz Parameters by the Laser Profilometer and Mitutoyo SJ 400

The workpiece surface quality values were measured with a laser profilometer on each workpiece in ten surface depths (Figure 8 right side). Namely at depths of 0.11; 2.31; 4.51; 6.71; 8.91; 11.11; 13.31; 15.51; 17.71; 19.91 mm. Material depths were the same for both measurement methods and materials.

The surface quality of the workpiece was measured by the Mitutoyo SJ 400 at four depths of the sample (Figure 8 left side). Namely at the depths of 0.11; 6.71; 13.31; 19.91 mm. The number of measurements has been reduced with respect to this method due to the complexity and difficulty of the measurement.

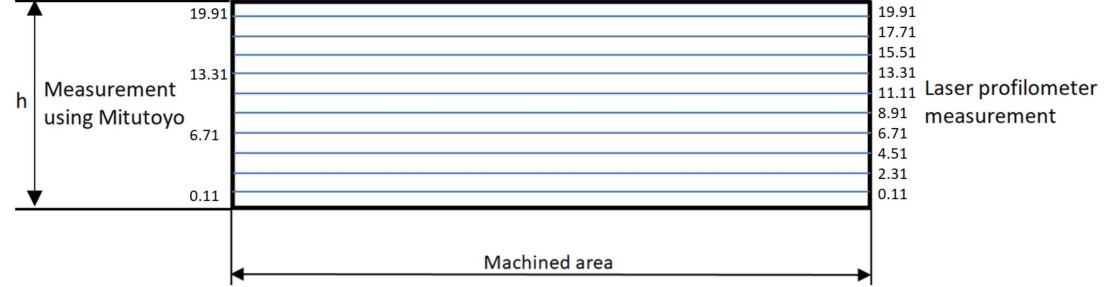

**Figure 8.** Sites measured by contactless method.

Table 5 shows the values measured with the Mitutoyo SJ400 roughness meter. These values are used to verify the values measured by the laser profilometer.

**Table 5.** Surface roughness of the samples measured by the Mitutoyo SJ400 at individual depths.

| Declared Roughness | Depth [mm] | EN5083 (Aluminium) | | A304 (Stainless-Steel) | |
|---|---|---|---|---|---|
| | | Ra [μm] | Rz [μm] | Ra [μm] | Rz [μm] |
| **6.3** | 0.11 | 2.74 | 16.2 | 2.38 | 14.3 |
| | 8.91 | 3.45 | 17.3 | 3.15 | 16.5 |
| | 15.51 | 4.82 | 20 | 4.22 | 17.9 |
| | 19.91 | 5.56 | 27.9 | 5.91 | 26 |

**Table 5.** *Cont.*

| Declared Roughness | Depth [mm] | EN5083 (Aluminium) | | A304 (Stainless-Steel) | |
|---|---|---|---|---|---|
| | | Ra [μm] | Rz [μm] | Ra [μm] | Rz [μm] |
| 12.5 | 0.11 | 3.10 | 21.4 | 3.71 | 19.2 |
| | 8.91 | 4.85 | 24.4 | 5.57 | 24.6 |
| | 15.51 | 6.82 | 34.2 | 8.94 | 39.5 |
| | 19.91 | 11.74 | 50.7 | 9.28 | 41.6 |
| 25 | 0.11 | 2.67 | 15.9 | 6.20 | 20.9 |
| | 8.91 | 5.65 | 30 | 8.94 | 41.8 |
| | 15.51 | 14.55 | 63.6 | 14.79 | 65.2 |
| | 19.91 | 15.99 | 67.7 | 21.67 | 75.1 |

## 3. Evaluation of the Measured Values

When the surface roughness was evaluated, the parameters of *Ra*—mean arithmetic deviation of the profile and those of *Rz*—the greatest height of the unevenness of the profile, were observed. The roughness of the surface was measured in 180 steps with a step size of 0.11 mm. The Gain (video signal strength gain) mode was set to 1. The shutter time (the exposure time that determines the time over which the surface is scanned) was 19.520 ms. After previous test measurements of this type of surface, these LPM device parameter setting values showed the clearest picture preview and the least noise in the image.

*Evaluation of Values Measured by Contactless Method*

This part of the paper describes the sample surface data measurement by the LPM system. Two graphic dependencies were created from the measured data for the *Ra* and *Rz* quality parameters, depending on the material of the measured area at given feed rates. The measured surfaces are examined at ten depths on three samples of stainless steel and three samples of aluminum. Parameters for setting the LPM input values for surface measurement are shown in Table 6. This behavior was in agreement with data observed by Valicek et al. [28] and Srivastava et al. [29].

**Table 6.** LPM setting parameters for sample measurement

| Axis Y Scan Distance | Axis X Scan Line |
|---|---|
| Scan lines count 200 μm | 4000 μm |
| Scan lines step 110 μm | |

The material samples surface roughness values measured were processed, resulting in two graphical dependencies *Ra* and *Rz*, for both materials examined. The results also confirm the conclusions of Jurko et al. [30] and Azhari et al. [31]:

Dependence of the mean arithmetic deviation of the *Ra* profile unevenness on the material depth h at the feed rate of AL120 mm·min$^{-1}$, AL220 mm·min$^{-1}$ and AL370 mm·min$^{-1}$ and SS050 mm·min$^{-1}$, SS120 mm·min$^{-1}$ and SS150 mm·min$^{-1}$,
Dependence of the greatest height of the *Rz* profile unevenness on the material depth h at the feed rate of AL120 mm·min$^{-1}$, AL220 mm·min$^{-1}$ and AL370 mm·min$^{-1}$ and SS050 mm·min$^{-1}$, SS120 mm·min$^{-1}$ and SS150 mm·min$^{-1}$.

The graphical dependence for the *Ra* parameter (Figure 9) shows the surface roughness in measured material depths at given feed rates v = 120, 220, 370 for aluminum and 50, 120 and 150 for stainless steel. This behavior was in agreement with data observed by Foldyna et al. [32] and Azhari et al. [33].

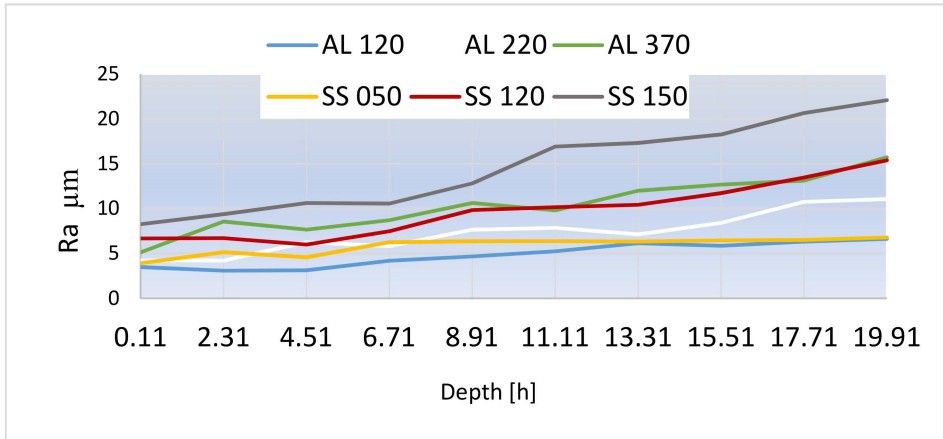

**Figure 9.** The graphical dependence of *Ra* on the measurement depth for AL120, AL220, AL370 and SS050, SS120, SS150.

The graphical dependence for the *Rz* parameter (Figure 10) shows the surface roughness in measured material depths at given feed rates v = 120, 220, 370 for aluminum and 50, 120 and 150 for stainless steel. This behavior was in agreement with data observed by Akkurt [34], Hreha et al. [35] and Boud et al. [36].

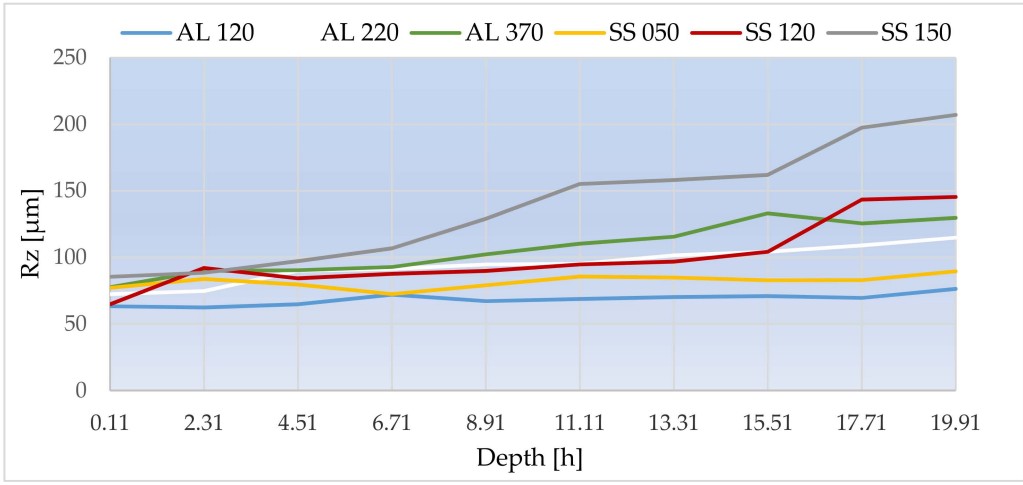

**Figure 10.** The graphical dependence of *Rz* on the measurement depth for AL120, AL220, AL370 and SS050, SS120, SS150.

In all samples, the curves of the *Ra* and *Rz* values have a rising tendency, depending on the increase in measured depth of the cut h.

Figure 11 illustrates 3D models of scanned samples made of aluminum, which was created by juxtaposing a series of profiles measured by LPM. 3D models show the surface area of the experimental samples captured with the laser profilometer. The images show a clear effect of cutting speed on the surface quality. The comparison of 3D models made (Figure 11, Figure 12) shows a clear difference in the surface quality of aluminum and stainless-steel samples. In samples made of aluminum (AL120, AL220, AL370), a smoother cut is evident. Smoother passing of the water jet did not leave as deep depressions on the surface of these samples as it did on the samples made of stainless steel. It follows from the comparison of graphical dependencies of the *Ra* and *Rz* parameters (Figure 9, Figure 10) that the roughness values are less jump-like, which reflects lesser strength and greater toughness of the evaluated material. This behavior was in agreement with data observed by Monno et al. [37] and Akkurt et al. [38].

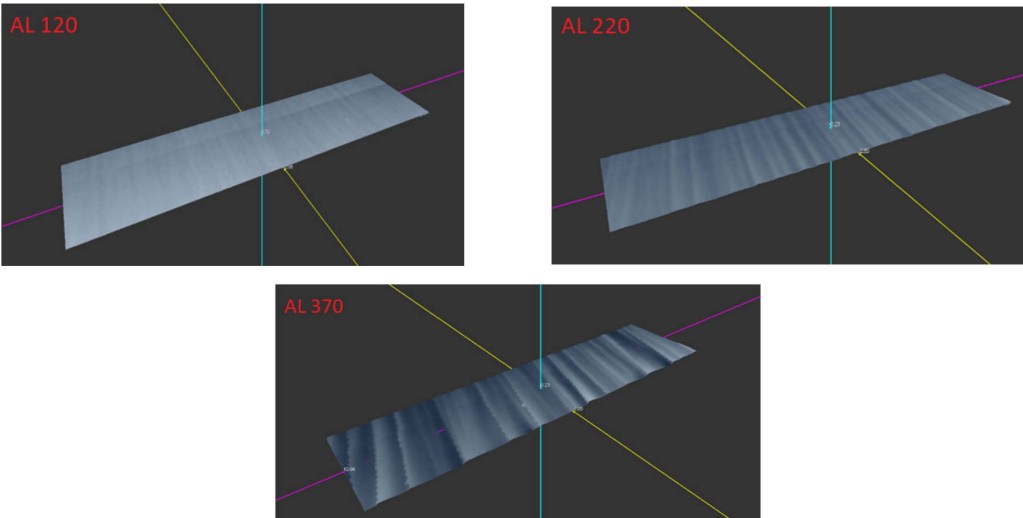

**Figure 11.** 3D model of the scanned EN5083 samples.

Figure 12 illustrates 3D models of the scanned stainless steel samples, created by juxtaposing a series of profiles measured by LPM. The effect of cutting speed on the quality of the stainless steel surface is shown here. In the experimental samples (SS050, SS120, SS150), greater difficulty of the water passing through the material is visible, with the water jet leaving more depressions on the sample surfaces. It follows from the comparison of graphical dependencies of the *Ra* and *Rz* parameters (Figure 9, Figure 10) that the roughness values are more jump-like, which reflects greater strength and lesser toughness of the material evaluated. Surface roughness in these experimental samples is greater in the smooth, medium and rough surface zones. In all samples, the curves of the *Ra* and *Rz* values have a rising tendency, depending on the increase in measured depth of the cut. This behavior is in agreement with data observed by Kunaporn et al. [39] and Hascalik et al. [40] and Tomas et al. [41].

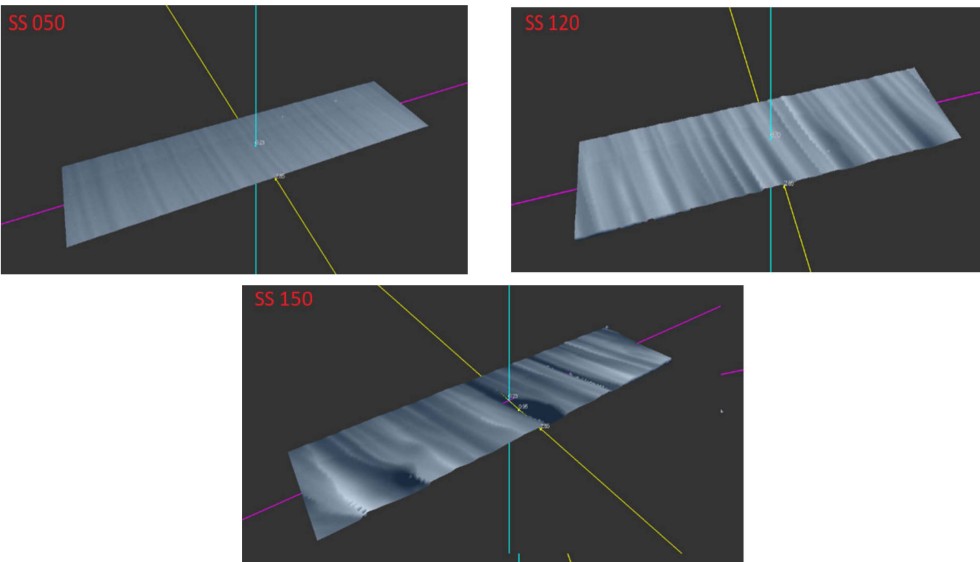

**Figure 12.** 3D model of the scanned A304 samples.

The Gain 1 mode measured the character of the surface in 180 step increments of 0.11 μm. The entire surface area of the sample was measured at ten measured sites when measured with LPM and four measured sites when measured with Mitutoyo SJ400. For comparison and verification of the measured values, samples made of aluminum and stainless steel were measured by the Mitutoyo SJ400

contact roughness tester (Table 5). This comparison shows that the *Ra* values, measured on the laser profilometer, lie within the range of the values measured by the Mitutoyo SJ 400. The *Rz* values are two to three times smaller than the values measured on the LPM, which, partly because of its geometry, is caused by the measuring tip of the contact roughness tester failing to enter every nook and cranny (the end of the measuring tip is larger than some surface depressions) and partly due to the noise in the LPM measurements, where in some steps a reflection of the CMOS camera occurred, resulting in a sharp increase in *Rz* values, i.e., measurement error. This measurement error can be removed by deleting the skewed values from exported tables.

## 4. Conclusions

The feed speed of the cutting head is one of the most important and technologically most easily adjustable technological parameters affecting the quality parameters Ra and Rz. Figures 9 and 10 show plotted dependencies of surface roughness *Ra*, *Rz* versus depth of material at the given cutting head feed rates of separating the respective materials. The measured and evaluated surface area of 0.11–19.91 μm gradually passed from smooth to rough surface zone (Figure 8).

Therefore, the measurement results can be summarized as follows:

- The plotted dependencies show that surface roughness varies linearly the deeper the AWJ cut.
- All plotted dependencies show a growing trend of roughness values.
- As the feed rate decreases, the smooth zone size (characterized by lower roughness values) increases.
- It occurs primarily in the first cut zone of the machined surface. This can also be seen in the plotted dependencies where the smallest initial surface roughness is (Figures 9 and 10).
- The values of the *Ra* and *Rz* parameters of the smooth zone are closest to each other, but even at these values, a difference in surface roughness is apparent, depending on the speed of the cutting head.
- The *Ra* and *Rz* parameter values of the AL120, AL220 and SS050, SS120 samples show a linear increase. The AL370 and SS150 samples exhibit a rather exponential increase in the *Ra* and *Rz* parameter values and their value trend is jump-like.
- Linearly increasing roughness is true for both the smooth and medium smooth zones (from 0.11 μm to 8.91 μm). An exponential increase in value occurs in the medium rough and rough (end) zone of the surface from 11.11 μm to 19.91 μm.
- It follows from the cuts and the measured values compared that the AWJ material separation technology shows equally manifesting increases. Occurrence of reliefs on the sectioned surfaces in both soft (AL) and hard metal materials (SS) was less frequent in the samples made of aluminum than in the samples made of stainless steel.
- This means that a different cutting head speed must be used to achieve the same surface roughness in different materials.

We can say that the roughness of the sample surfaces produced by the DRC Company did not exceed the declared roughness of any of the samples used. The task of the experiment was to compare and evaluate the surface of samples produced by the AWJ technology at the DRC Company and to evaluate the measurement methods. The company declares that the surface roughness of each sample produced is not greater than that shown in Table 2 for each of the samples. The values measured with the contact roughness meter Mitutoyo SJ 400 show its first obvious disadvantage compared to the LPM device. The contact roughness meter does not provide for a complete picture of the surface under evaluation or the roughness of its individual parts. This is because, in measuring the 800 μm section of the surface (the section covered by the tip of the instrument over the measured surface), the result is a single average value of the roughness parameters *Ra* and *Rz*. When measured with LPM, the measurement result is a set of values that describe the surface areas at each measurement step.

The LPM also evaluates the largest depression and the largest protrusion of the surface under evaluation, i.e., it evaluates several parameters in the individual measurement steps. Another advantage is the creation of a 3D image of the examined surface where it is possible to better see the surface nature of the object being measured.

**Author Contributions:** G.M. suggested the concept of application of laser profilometry to evaluation of the nature of the surface of the workpiece separated by the advanced water jet technology; Š.O. prepared testing samples; G.M. and J.R. designed and performed experimental measurements in the laboratory; J.D. and Š.O. analyzed and evaluated measured data; G.M. contributed materials and analysis tools; J.D. wrote the paper.

**Acknowledgments:** This research work was supported by project KEGA no. 006TUKE-4/2017 "Innovation of Laboratory quality control of components for the automotive and allied industries within the framework of the integration of advanced cognitive operations into education". We thank unknown reviewers for valuable recommendations of text improvement.

**Conflicts of Interest:** The authors declare no conflict of interest. The founding sponsors had no role in the design of the study; in the collection, analyses, or interpretation of data; in the writing of the manuscript, and in the decision to publish the results.

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
