# Peer review of "Application of Laser Profilometry to Evaluation of the Surface of the Workpiece Machined by Abrasive Waterjet Technology"

_applsci, doi:10.3390/app9102134_

Round 1

Reviewer 1 Report

Topic of reviewed article is very important especially in context of complementary evaluation of repeatability of the machining results in abrasive water-jet processes. Reviewed article is interesting but presents low scientific quality. Week points mentioned below should be clarify and improved before publication:

word “nature” in title did not reflect the content of the text – title should be redefined and write more precise,

lack of description of scientific problem in the text,

the goal of this study should be clearly given at the end of introduction section and this goal should result from the literature analysis,

some recent articles concerning laser measurements of surface texture should be add to the literature review,

all symbols should be consequently write italics and all acronyms should be described with the first use in the text,

why authors provide description of technology (AWJ) in methodology (section 2.1) – in this place should be given specific information about machinery, materials and methods used in the study, not general description of benefits (on top of it without giving the drawbacks and restrictions),

please clarify the motivation of selection of machined materials,

Fig. 4 have two parts but description did not define content of left and right photo (the same Fig. 8, 9, 15 and 18),

what does it mean “LPM Laser Profilometer” – please be more precise with description of all figures,

Fig. 10 has very poor quality,

why once authors name measurement method as “laser profilometry” and then use term “contactless method” – this terms have different meaning (please consequently use proper term),

language should be improved,

dot is not a correct symbol of multiplication (mm.min),

what is the point of giving tables with values (i.e. tab. 5-7) and then chart (fig. 13) – all values should be given on the chart and tables should be removed,

why authors provide Fig. 15 and 18 without any comment?

deeper scientific analysis of obtained results should be provided,

conclusions are mainly write as abstract, which they aren’t – entire section should be rewrite,

main conclusions should be write as numbered list,

in conclusions lack of scientific explanation of observed phenomena and trends.

Author Response

Response to Reviewer 1 Comments

Point 1: Word “nature” in title did not reflect the content of the text – title should be redefined and write more precise.

Response 1: Article title was edited.

Point 2: Lack of description of scientific problem in the text.

Response 2: The description of the scientific problem has been edited (Introduction).

Point 3: The goal of this study should be clearly given at the end of introduction section and this goal should result from the literature analysis.

Response 3: The goal of the study was mentioned at the end of Introduction.

Point 4: Some recent articles concerning laser measurements of surface texture should be add to the literature review.

Response 4: Recent articles concerning laser measurement of surface texture was added to the literature review.

Point 5: All symbols should be consequently write italics and all acronyms should be described with the first use in the text.

Response 5: All symbols and acronyms have been modified in the text.

Point 6: Why authors provide description of technology (AWJ) in methodology (section 2.1) – in this place should be given specific information about machinery, materials and methods used in the study, not general description of benefits (on top of it without giving the drawbacks and restrictions).

Response 6: Section 2.1 was edited according to your reviews.

Point 7: Please clarify the motivation of selection of machined materials.

Response 7: The selection of machined materials has been explained in the text.

Point 8: Fig. 4 have two parts but description did not define content of left and right photo (the same Fig. 8, 9, 15 and 18).

Response 8: Figures were edited.

Point 9: What does it mean “LPM Laser Profilometer” – please be more precise with description of all figures.

Response 9: LPM Laser Profilometer – device used for contactless method. It is described in detail in chapter 2.2.

Point 10: Fig. 10 has very poor quality.

Response 10: Figure 10 has been modified.

Point 11: why once authors name measurement method as “laser profilometry” and then use term “contactless method” – this terms have different meaning (please consequently use proper term).

Response 11: The correct term has been modified throughout the article.

Point 12: Language should be improved.

Response 12: Please see attached Certificate of proofreading.

Point 13: Dot is not a correct symbol of multiplication (mm.min).

Response 13: Mathematical expressions have been modified.

Point 14: What is the point of giving tables with values (i.e. tab. 5-7) and then chart (fig. 13) – all values should be given on the chart and tables should be removed.

Response 14: Tables were removed from the text.

Point 15: Why authors provide Fig. 15 and 18 without any comment.

Response 15: Comment on image has been added.

Point 16: Deeper scientific analysis of obtained results should be provided.

Response 16: A deeper scientific analysis of the results achieved has been developed.

Point 17: Conclusions are mainly write as abstract, which they aren’t – entire section should be rewrite.

Response 17: The conclusion was revised.

Point 18: Main conclusions should be write as numbered list.

Response 18: The conclusion was revised.

Point 19: In conclusions lack of scientific explanation of observed phenomena and trends.

Response 19: The conclusion was revised.

Reviewer 2 Report

Row 47 - 48: Reference of the names Z. Cojbasic, Petkovic,
48 Shamshirband, Tong, Ch, Jankovic, Ducic, Baralic
is carried out without the corresponding reference in the literature

Row 83: It's not so good to start the sub-chapter at the bottom of the page without using text

Row 114: Figure 3 title must be with the figure on the same page. The figure 's title does not refer to the samples.
It is the same with the Figure 2 title.

Row 123 - 129: First must be given the definitions and then Tables 1 and 2 (row 130 and 131).

Row 138: Figure 4 title must be with the figure on the same page.

Row 198: It's not so good to start the sub-chapter at the bottom of the page without using text

Row 240: Figure 13 title must be with the figure on the same page.

Author Response

Response to Reviewer 2 Comments

Point 1: Row 47 - 48: Reference of the names Z. Cojbasic, Petkovic, 48 Shamshirband, Tong, Ch, Jankovic, Ducic, Baralic is carried out without the corresponding reference in the literature.

Response 1: Number of reference has been inserted in the text.

Point 2: Row 83: It's not so good to start the sub-chapter at the bottom of the page without using text.

Response 2: Text has been edited.

Point 3: Row 114: Figure 3 title must be with the figure on the same page. The figure 's title does not refer to the samples. It is the same with the Figure 2 title.

Response 3: Figures has been edited.

Point 4: Row 123 - 129: First must be given the definitions and then Tables 1 and 2 (row 130 and 131).

Response 4: Definition is placed before the tables.

Point 5: Row 138: Figure 4 title must be with the figure on the same page.

Response 5: Edited.

Point 6: Row 198: It's not so good to start the sub-chapter at the bottom of the page without using text.

Response 6: Edited.

Point 7: Row 240: Figure 13 title must be with the figure on the same page.

Response 7: Edited.

Reviewer 3 Report

The authors should provide a table summarising previous work on similar topics listing the type of material used and its thickness, cutting parameters used , what was measured and the main findings. at least ten references to include in the table

using acronym in the title is not recommended, please consider changing it or modifying the title

abstract needs corrections, the authors should clearly explain that they are comparing the surface roughness measurements of metallic samples using 3D and 2D profilometry equipment. 

no need to mention the roughness devices model numbers and names in the abstract

Line 26 "different feed rate of the cutting head was used" should be different feed rates of the cutting head were used! please double check

Consider defining Rz properly it is the ten as it is the arithmetic mean value of the single roughness depths of consecutive sampling lengths.

Figure 1 is not clear, please improve its quality

Figure 2 is not necessary, just showing a photo of the machine is not enough, it is advised for the authors to show more details of the machine or remove the figure

Figure 4 please add details with arrows describing what are we looking at

Figure 6 is not clear, it is not clear what the authors are trying to show us here. if just showing the software capabilities then it is not necessary to show it as screenshots in a figure. it is better to describe the process and what were the parameters used within the text content

Figure 7, 8 and 9,12,15 and 18: please see comments from Figure 6, add arrows and show the readers what are we looking for in the image.

combine Figures 10 and 11

combine figure 13 and 14

combine tables 8-10

combine figures 16 and 17

conclusion is very long, the authors should reduce it and only add main findings which you would like to tell the readers from the study.

Author Response

Response to Reviewer 3 Comments

Point 1: The authors should provide a table summarising previous work on similar topics listing the type of material used and its thickness, cutting parameters used, what was measured and the main findings. At least ten references to include in the table.

Response 1: Previous work on similar topics is mentioned in the evaluation (12 references). It is very difficult to indicate in the table the thickness, cutting parameters used, what was measured and so on. We hope it can be that way.

Point 2: Using acronym in the title is not recommended, please consider changing it or modifying the title.

Response 2: Acronym has been removed from the title.

Point 3: Abstract needs corrections, the authors should clearly explain that they are comparing the surface roughness measurements of metallic samples using 3D and 2D profilometry equipment.

Response 3: Abstract has been edited.

Point 4: No need to mention the roughness devices model numbers and names in the abstract.

Response 4: Abstract has been edited.

Point 5: Line 26 "different feed rate of the cutting head was used" should be different feed rates of the cutting head were used! Please double check.

Response 5: Thank you for your caution. Abstract has been edited.

Point 6: Consider defining Rz properly it is the ten as it is the arithmetic mean value of the single roughness depths of consecutive sampling lengths.

Response 6: Thank you for your explanation. Abstract has been edited.

Point 7: Figure 1 is not clear, please improve its quality.

Response 7: Figure 1 has been edited.

Point 8: Figure 2 is not necessary, just showing a photo of the machine is not enough, it is advised for the authors to show more details of the machine or remove the figure.

Response 8: Technical parameters of the machine has been added (see table 1).

Point 9: Figure 4 please add details with arrows describing what are we looking at.

Response 9: Figure 4 has been edited.

Point 10: Figure 6 is not clear, it is not clear what the authors are trying to show us here. If just showing the software capabilities then it is not necessary to show it as screenshots in a figure. It is better to describe the process and what were the parameters used within the text content.

Response 10: Figure 6 has been removed.

Point 11: Figure 7, 8 and 9,12,15 and 18: please see comments from Figure 6, add arrows and show the readers what are we looking for in the image.

Response 11: Figures were edited according to your instructions.

Point 12: Combine Figures 10 and 11.

Response 12: Figures were combined.

Point 13: Combine figure 13 and 14.

Response 13: We cannot combine different roughness parameters (Ra and Rz). We combined figures 13,16 and 14,17.

Point 14: Combine tables 8-10.

Response 14: Tables 8-10 were removed from the paper according to reviewer 1 comments.

Point 15: Combine figures 16 and 17.

Response 15: We cannot combine different roughness parameters (Ra and Rz). We combined figures 13,16 and 14,17.

Point 16: conclusion is very long, the authors should reduce it and only add main findings which you would like to tell the readers from the study.

Response 16: We cannot combine different roughness parameters (Ra and Rz). We combined figures 13,16 and 14,17.

Round 2

Reviewer 1 Report

All provided changes in the text should be highlited.

In current form it is very hard to identifiy provided changes, especially when response for given comments are very curt.

Author Response

Point 1: All provided changes in the text should be highlighted.

In current form it is very hard to identify provided changes, especially when response for given comments are very curt.

Response 1: All provided changes in the text was clearly highlighted using “Track Changes” function in Microsoft Word.

Reviewer 3 Report

The authours have answerd all quereies and the paper can be accepted for publication

Author Response

Reviewer statement (see below Reviewer report form):

The authours have answerd all quereies and the paper can be accepted for publication

Round 3

Reviewer 1 Report

All my comments were taken into consideration.

This text is suitable for publication in current form.